# What I don't know can hurt you: Collateral combat damage seems more acceptable when bystander victims are unidentified

Scott Danielson[1], Paul Conway[2]*, Andrew Vonasch[1]

1 Department of Psychology, Speech and Hearing, University of Canterbury, Christchurch, New Zealand,
2 Centre for Research on Self and Identity, School of Psychology, University of Southampton, Southampton, United Kingdom

* P.Conway@soton.ac.uk

**Data Availability Statement:** Datasets and code can be found at: https://osf.io/c5hjq/?view_only=71191a9d5fe14c6fbf548418d1a04a51.

## Abstract

Five experiments ($N$ = 2,204) examined responses to a realistic moral dilemma: a military pilot must decide whether to bomb a dangerous enemy target, also killing a bystander. Few people endorsed bombing when the bystander was an innocent civilian; however, when the bystander's identity was unknown, over twice as many people endorsed the bombing. Follow-up studies tested boundary conditions and found the effect to extend beyond modern-day conflicts in the Middle East, showing a similar pattern of judgment for a fictional war. Bombing endorsement was predicted by attitudes towards *total war*, the theory that there should be no distinction between military and civilian targets in wartime conflict. Bombing endorsement was lower for UK compared to US participants due to differences in total war attitudes. This work has implications for conflicts where unidentified bystanders are common by revealing a potentially deadly bias: people often assume unidentified bystanders are guilty unless proven innocent.

## Introduction

About as many civilians as soldiers die in war each year [1, 2], some during strikes targeted at enemy combatants. For instance, US airstrikes targeting ISIS militants reportedly caused a steep civilian death toll across multiple incidents [3, 4]. Some of these may be calculated sacrifices, but there have been many reported cases of mistaking innocent civilians for enemy combatants, with the possibility of many more being unreported [4, 5]. Why do military strikes so often strike innocent non-combatants? The current research offers five studies testing a potential explanation why: people tend to assume unknown bystanders in a combat zone are enemies rather than civilians, reducing concerns about collateral damage.

Many factors influence sacrificial decisions, such as emotional impact of harm, number of lives saved, and self-presentation [6–10]. Although most dilemma research ignores social relationships [11, 12], sacrifices increase when targets appear different from decision-makers [13] or evil and blameworthy [14]. Accordingly, war increases willingness to sacrifice outgroups to achieve utilitarian outcomes [15].

**Funding:** The author(s) received no specific funding for this work.

**Competing interests:** The authors have declared that no competing interests exist.

In war, danger is prevalent. People are generally sensitive to threats [16] and bad intentions [17], but wartime may amplify such processing. Threatening environments increase detection of harmful agents [18], categorization of others as outgroups [19], and perception of outgroups as threatening [20] and untrustworthy [21]. Hence, wartime threats may lead people to assume that unknown targets are enemies, justifying attack.

Moreover, war entails uncertainty. Theoretically, uncertainty could either amplify or decrease hesitation to sacrifice. For example, sacrificing one person who *might* die to save five who *might* live seems worse than definitely sacrificing one to save five [22, 23]. On the other hand, uncertainty can also absolve deciders of responsibility for their decisions [24], suggesting more unknowns in war might make wartime killing seem less risky. For instance, people may be more willing to sacrifice bystanders of uncertain identities, due to the possibility they are enemies. Prospect theory agues uncertainty can be appealing or unappealing depending on whether a decision-maker is trying to minimize losses or maximize gains [25]. Loss consideration might be salient in war. For instance, support for the Iraq War waned as news reported more deaths of US troops, with public sentiment most impacted when those troops were from local areas [26]. However, wartime conflict could also be framed in terms of strategic gains, like benefits to the self or to the group, or punishing those we perceive to be morally wrong [27]. Given these two perspectives people may take on war, it is unclear what prospect theory would predict in terms of wartime judgments involving uncertainty. Taken together, this work shows that uncertainty can impact judgments, especially concerning losses, moral elements, and utilitarian choices. Despite a large body of evidence and theory however, it remains unclear what these findings predict for wartime moral judgments.

Killing non-combatants in an enemy nation during war is not always considered a sacrifice. *Total war* approaches to combat conceptualize civilians as legitimate military targets [28, 29] because they conceptualize the struggle as between entire nations. This thinking was common during the Second World War and despite it being less common in modern warfare, it is still present today. Belief in total war has not been widely studied as a psychological construct. In fact, at the time of writing, belief in total war appears to have been studied not at all in empirical psychology, yet it could have large downstream impacts on people's judgment of what is ethical and expected in war. People who endorse total war principles may view enemy civilian deaths as acceptable or even desirable, compared to people who instead favor *limited warfare* between military combatants only. For supporters of total war, civilian deaths are not seen as a sacrifice or an error in judgments, but an inevitable feature of war.

To understand real world behavior, it is important to study situations similar to the real world. Wartime decision-making is a context where crucial moral decisions are made, yet most of the work in the psychology of moral tradeoffs and sacrifice has not been done in this context [6–14]. There a significant gap in the literature concerning judgments of the kind of moral trade-offs made in the military armed conflicts of today, particularly ones involving a realistic level of uncertainty. Therefore, across five studies, (*N* = 2204), we investigated to what extent people would endorse a military bombing that would as a side effect kill an unidentified bystander. Participants considered a dilemma set in the real-world conflict between the United States and Islamic State (ISIS) (S1-S3 Tables in S1 File for full text). Studies 1 and 2 explored the extent to which participants endorsed bombing which would kill an unidentified bystander. Using experimental vignettes, we examined how a lack of information about a bystander impacts an individual's endorsement of the bombing as compared to a known bystander (innocent or otherwise) or non-innocent combatant. We also examine whether participants given no information about a wartime bystander make any assumptions about their allegiance, i.e., whether they are combatants or civilians. In Study 3 we extended this paradigm outside the real-world setting of the ISIS conflict into a fictional conflict, where individual

judgments would not favor one side of the conflict over another. Study 4 investigated how attitudes toward total war impact endorsement of bombing civilian bystanders, and Study 5 replicated this and Study 1 among UK, rather than US, participants. Study 1 was exploratory; we preregistered Studies 2–5.

Studies were approved by the Florida State University Human Subjects Ethics Committee, HSC No. 2016.18130 and The University of Canterbury Human Research Ethics Committee, HEC 2020/16/LR-PS. We obtained consent for all studies by asking participants to select a mandatory consent item before proceeding with the study. The dates of running each study were as follows: October 1–2, 2016, Study 1b: May 31- June 3 2020, Study 1c: January 14 2021, Study 2: September 17 2020, Study 3: August 17–20, 2020, Study 4: February 28, 2022, Study 5: April 11, 2022, Supplemental Study 1: March 18–19, 2021. All materials are reported between the main text and supplemental materials (see S1 File). Datasets and code can be found at: [https://osf.io/c5hjq/?view_only=71191a9d5fe14c6fbf548418d1a04a51].

## Study 1a

Study 1a examined whether the identity of a sacrificial bystander influences decisions to bomb a military target. We compare how many participants endorse a military strike that would kill an unknown bystander to a strike that would kill an either innocent bystander, another combatant, or one of several other bystanders tied in some way to the enemy.

### Method

In 2016, we recruited 464 adult Americans via Amazon's Mechanical Turk (MTurk). Responses included 194 men (42%), 269 women (58%) and 1 other (>1%), and ranged in age between 18 and 74 years ($M = 36.72$, $SD = 12.01$). We analyzed all completed responses; no exclusions. The survey was open to any user on the online platform and had no screening questions; this was also the case for all future studies. In a sensitivity analysis, the pwr package [30] for R Statistics revealed 80% power to detect effect sizes of Cohen's $w = 0.17$. We report demographics for all studies in Table 1. Participants read about a US pilot in Iraq who must choose whether to a) fire a missile on a farmhouse where there was an ISIS combatant known for making chemical weapons and bystander thus killing them both, or b) refuse to bomb, letting both individuals live (S1 Table in S1 File for full text).

We varied the bystander's identity across seven conditions. We manipulated the bystander's combatant status and degree of support and involvement in the conflict. We described the bystander as either a) another ISIS soldier, b) a devout Muslim who agrees with ISIS's extremism, c) a civilian farmer who profits off the war, d) a devout Muslim who feels conflicted about extremism, e) a devout Muslim who rejects extremism, f) an innocent civilian farmer, or g) an unidentified person about whom nothing is known. These conditions aim to create a spectrum of involvement ranging between an enemy combatant and an uninvolved civilian bystander.

**Table 1. Sample sizes and demographics for all studies.**

| Study | Study Manipulation | N | Nationality | Mean Age (SD) | Gender (% Female) |
|---|---|---|---|---|---|
| 1a | Bystander's association to the enemy target (7 levels) | 464 | United States | 36.70 (12.00) | 57.97% (0.28% other) |
| 1b | Bystander's association to the enemy target (7 levels) | 271 | United States | 30.90 (10.78) | 47.23% (0.00% other) |
| 1c | Unidentified bystander only | 93 | United States | 31.23 (11.19) | 52.68% (0.02% other) |
| 2 | Percent chance the bystander is an enemy combatant (11 levels) | 297 | United States | 37.76 (11.97) | 41.75% (1.01% other) |
| 3 | Real world wars versus fictional wars (2 levels) | 203 | United States | 33.15 (11.29) | 59.70% (1.00% other) |
| 4 | Foreign/local and innocent/guilty bystander; total war attitudes | 579 | United States | 38.4 (14.4) | 50.26% (0.52% other) |
| 5 | UK replication of Study 1 (7 levels) | 297 | United Kingdom | 37.99 (13.22) | 49.83% (0.68% other) |

To measure bombing endorsement participants decided, "Should the pilot fire upon the building to kill both people inside?" (*Yes/No*) and "How acceptable is it for the pilot to fire upon the building, killing both people inside?" on a scale from 1 (*Not at all*) to 7 (*Very much*). We predicted that people would endorse sacrificing the innocent civilian bystander less compared to another enemy combatant, and all other conditions would fall between these extremes. However, participants may endorse the bombing wholeheartedly or reluctantly. To assess this aspect of the dilemma we included three questions: "How comfortable are you with your decision?" (1–7, *Not very-Very*), "How much does the ISIS Operative who makes chemical weapons deserve to die?" (1–7, *Not at all-Very much*), and "How much does the second person who was already in the farmhouse deserve to die? (1–7, *Not at all-Very much*).

## Results

We anticipated that people would be sensitive to the identity of a bystander in the line of fire when judging a potential military strike. As predicted, the bystander's identity significantly impacted endorsement of bombing for the binary yes/no question, $\chi^2(6,464) = 100.5$, $p < .001$, $w = 0.47$ (Fig 1). The Likert style question which asked about the acceptability of firing also showed a significant impact of bystander identity with a similar pattern of results in a non-parametric Kruskal-Wallis ANOVA, $\chi^2(6,456) = 84.98$, p < .001, $\varepsilon^2 = 0.18$ (S1 Fig in S1 File). In the binary yes/no question people were most willing to bomb when the bystander was an enemy (77.9%) and least willing (24.2%) when they were an innocent civilian. Willingness to bomb decreased relative to how much the bystander was involved in the conflict. For example, 53.7% agreed to bomb a Muslim extremist who supported ISIS and only 31.8% endorsed the bombing when the bystander was a farmer who benefitted financially from the ISIS takeover.

Unexpectedly, over half of participants (58.5%) endorsed bombing unidentified bystanders despite no concrete evidence they were enemies. This was roughly twice the bombing rate of civilian bystanders, similar to bombing Muslim extremists. Bonferroni corrected pairwise comparisons in a generalized linear model using a log link transform showed that people were

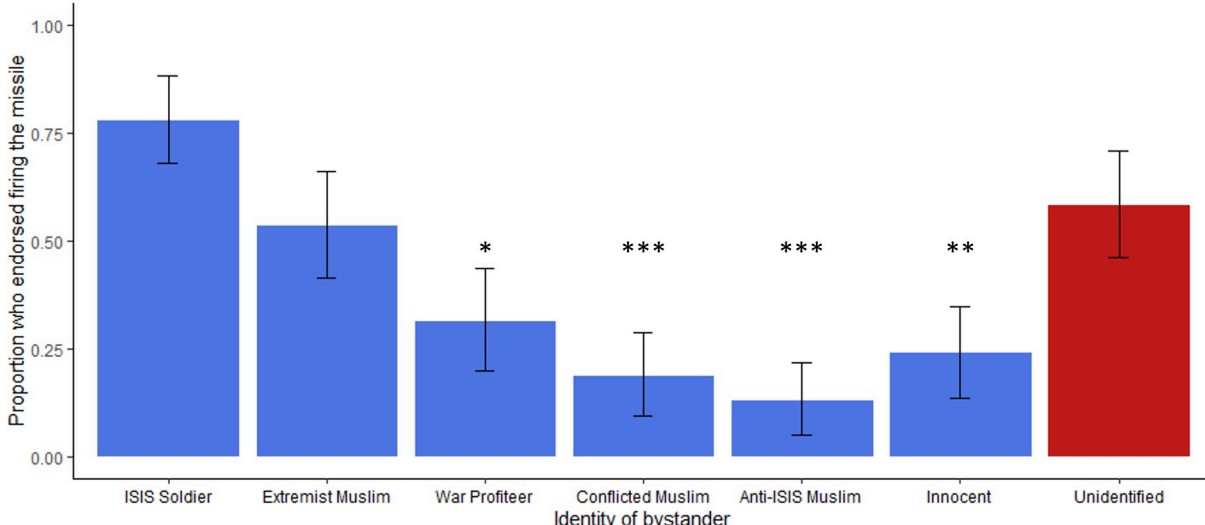

**Fig 1. Percent of "Yes" responses to "Should the pilot fire?" depending on bystander identity, Study 1a.** Error bars reflect 95% CIs. Asterisks denote Bonferroni corrected p-values compared with the unidentified bystander condition (*$p < .05$, **$p < .01$, ***$p < .001$). More innocent and uninvolved bystanders see a reduction in firing compared to bystanders who were guilty (e.g., ISIS solider) or tied to ISIS (e.g., Extremist Muslim, civilian war profiteer). This pattern does not hold for the unidentified bystander condition, for which most (58.5%) participants said the pilot should fire.

significantly more willing to bomb an unidentified than innocent civilian ($exp(B)$ = 1.48, $SE$ = .382, $p$ = .002), anti-ISIS Muslim ($exp(B)$ = 2.21, $SE$ = .438, $p$ < .001), a Muslim conflicted about ISIS ($exp(B)$ = 1.78, $SE$ = .398, $p$ < .001), or a war profiteer ($exp(B)$ = 1.11, $SE$ = .085, $p$ = .058. Conversely, unidentified bystander bombing rates did not significantly differ from an ISIS soldier ($exp(B)$ = -.920, $SE$ = .386, $p$ = .358), or the Muslim extremist ($exp(B)$ = .192, $SE$ = .351, $p$ = 1.00, Fig 1). There was no impact of gender on whether participants were likely to endorse firing, $\chi^2(1, 463)$ = 1.11, $p$ = .292, $OR$ = 0.82, 95% CI[0.56,1.19]). One participant who marked "other" for gender was excluded from this analysis due to small sample size. Similarly there was no effect of age on firing endorsement when included as a covariate, $\chi^2(1,464)$ = 5.27, $p$ = .311, $OR$ = 1.01.

Several more items assessed the manipulation of the different bystander identities. When participants were asked how comfortable they were with their decision, there was no impact of condition in a non-parametric Kruskal-Wallis one way ANOVA, $\chi^2(6)$ = 10.08, $p$ = .121, $\varepsilon^2$ = 0.02. However, an exploratory analysis found an effect of participant's decisions to endorse bombing: those who opted not to bomb were significantly more comfortable with their decision overall ($M$ = 5.38, $SD$ = 1.75) than those who opted to bomb ($M$ = 4.63, $SD$ = 1.96), U($N_{Endorse}$ = 185, $N_{Not\ endorse}$ = 279) = 20085, $p$ < .001, $r$ = 0.22. Participants were also asked in separate questions how much each person—the targeted ISIS fighter and the bystander—deserved to die. Participants showed high agreement that the ISIS fighter deserved to die across conditions ($M$ = 5.08, $SD$ = 1.87) with no difference between conditions which changed the bystander, $\chi^2(6)$ = 2.38, $p$ = .881, $\varepsilon^2$ = 0.01. However, ratings of bystander's deservingness to die varied significantly across the different bystander identities, $\chi^2(6)$ = 173.46, $p$ < .001, $\varepsilon^2$ = 0.38. This broadly followed the same pattern as firing acceptability and firing endorsement (Fig 2). However, participants rated the unidentified bystander as significantly more deserving of death than any other non-combatant bystanders—even the ISIS sympathizer and war profiteer—according to a Mann-Whitney test, U($N_{non-combatants}$ = 330, $N_{unidentified}$ = 65) = 5040.50,

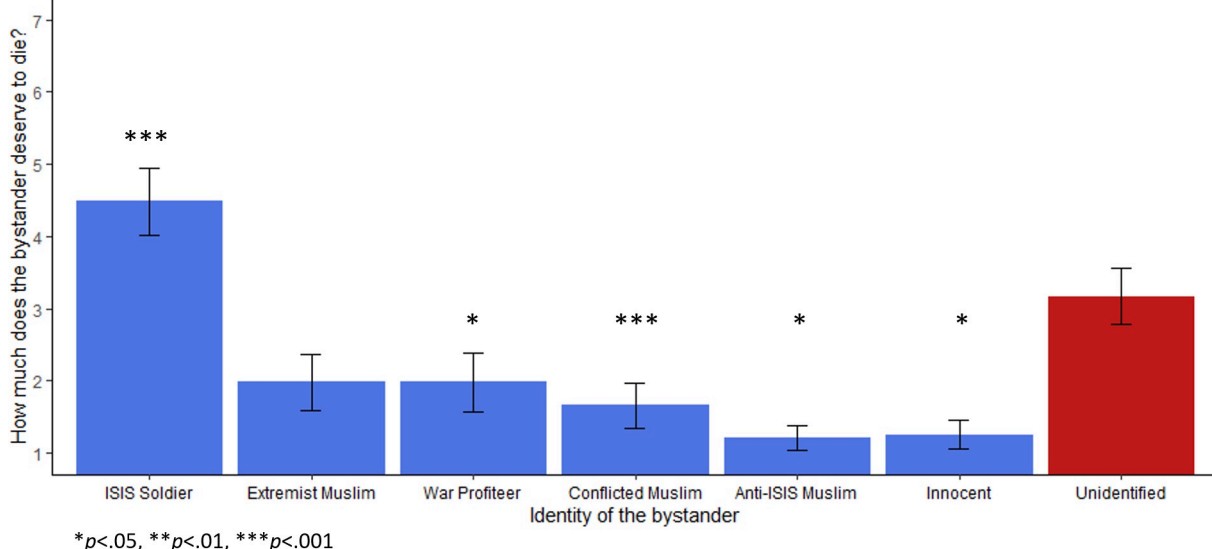

*p<.05, **p<.01, ***p<.001

**Fig 2. Rating of bystander deservingness to die depending on identity, Study 1a.** Error bars represent a 95% confidence interval. Asterisks denote Games-Howell corrected p-values compared with the unidentified bystander condition (*$p$ < .05, **$p$ < .01, ***$p$ < .001). Participants answered the question "How much does the second person [the bystander] deserve to die?" with a Likert scale between 1 (*Not at all*) and 7 (*Very much*). The bystander rated most deserving of death was the ISIS soldier ($M$ = 4.49, $SD$ = 1.93), the second highest was the unidentified bystander ($M$ = 3.1, $SD$ = 1.55).

$p < .001$, $M_{Diff}$ = -2.00, 95%CI[-3.00, -2.00], rank biserial correlation = 0.53 (coded unidentified bystander 1, other civilian bystanders 0). However, participants rated the unknown bystander as less deserving than the ISIS soldier bystander, $U(N_{Soldier} = 68, N_{unidentified} = 65) = 1274.00$, $p < .001$, $M_{Diff}$ = -1.00, 95%CI[1.00, 2.00], biserial rank correlation = 0.42. This suggests that not only do many participants endorse a bombing that would kill an unidentified bystander, many also judge that the unidentified bystander deserved it.

## Discussion

To kill a dangerous enemy, people were more willing to sacrifice an unknown bystander than any known bystander except when the bystander was an enemy combatant or Muslim extremist. Likewise, people rated the unknown bystander more deserving of death than any other bystander except an enemy combatant. These results suggest people treated unidentified bystanders more like enemy combatants than innocent (or even morally compromised) civilians.

## Study 1b

Study 1a surveyed participants in 2016 during the height of the US-ISIS conflict. Shifting public sentiment about this conflict may limit its generalizability. Study 1b replicated Study 1a in 2020 after much of the threat and news coverage of the ISIS conflict had abated. Preregistered: https://aspredicted.org/EJU_BYG.

## Method

We recruited 317 American participants from mTurk. After excluding incomplete responses and participants who failed a competence check we were left with a final sample of 271. The study included 143 (53%) males and 128 (47%) females and ranged in age between 18 and 70 ($M = 30.90$ $SD = 10.78$). A sensitivity analysis showed 80% power to detect effect sizes of Cohen's $w = 0.22$. We presented the same materials and measures as study 1a and added an item asking participants to explain their reasoning for their decision about the bombing in a free response format.

## Results

As in Study 1a which was run in 2016, the identity of the bystander again mattered for endorsing the bombing, $\chi^2(6) = 49.42$, $p < .001$. As in study 1a there was no effect of gender, $\chi^2(1,271) = 0.17$, $p = .682$, $OR = 0.90$, 95%CI[0.55,1.49], nor age, $\chi^2(1,271) = 3.18$, $p = .075$, $OR = 0.98$, 95% CI[0.96,1.00]. The patterns of results for endorsing the bombing were similar to Study 1a (Fig 3). When both datasets were combined and study was included as a factor along with condition to predict bombing endorsement, there was no main effect of study, $\chi^2(1) = 1.78$, $p = .183$, exp(B) = 0.79, but a small significant interaction between condition and study, $\chi^2(6) = 14.43$, $p = .025$. However, no Bonferroni corrected pairwise comparisons between conditions across studies 1a and 1b reached significance. The greatest difference was in the Muslim extremist condition, which saw less endorsement of firing in 2020 (23.1%) than it did in 2016 (*48.1%*), but nevertheless the difference was nonsignificant, exp(B) = 3.87, $p = .251$. Additionally, as in Study 1a, ratings of acceptability of the bombing showed a significant effect of condition in a Kruskal-Wallis ANOVA, $\chi^2(6, 264) = 54.91$, $p < .001$, $\varepsilon^2 = 0.20$. The pattern of results was also similar to that of Study 1a (Fig 3).

Participants were asked how comfortable they were with their decision, which unlike study 1a showed a medium effect of condition in a Kruskal-Wallis ANOVA, $\chi^2(6) = 16.62$, $p = .011$, $\varepsilon^2 = 0.06$. Pairwise comparisons showed significant differences only in the conflicted Muslim

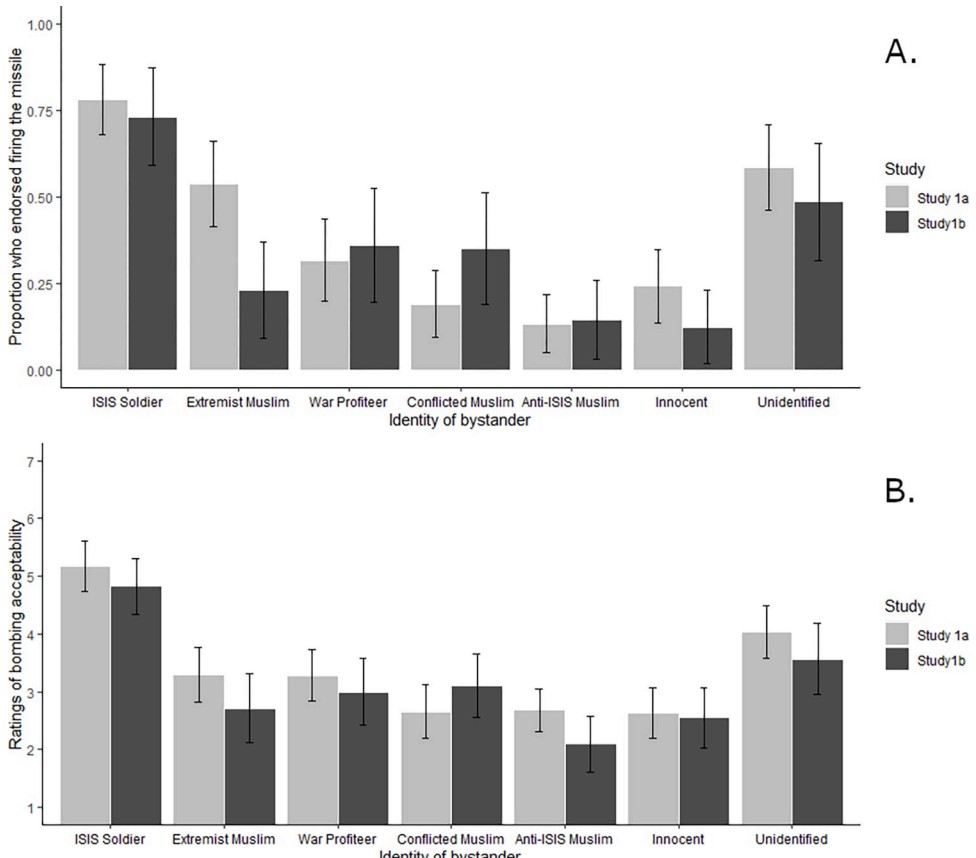

**Fig 3.** A) Percent of "Yes" Responses to "Should The Pilot Fire?" and B) Rating of firing acceptability Depending on Bystander Identity in 2016 (Study 1a) and 2020 (Study 1b). Error bars represent a 95% confidence interval. A) Bonferroni-corrected post-hoc comparisons showed no significant differences in firing endorsement by study for any condition. B) Dwass-Steel-Critchlow-Fligner pairwise comparisons showed no significant differences in bombing acceptability by study for any condition.

condition, in which participants rated their comfort as significantly lower than participants in the Innocent condition ($W = 4.78$, $p = .013$) and the anti-ISIS Muslim condition ($W = 4.46$, $p = .027$). However, this effect is likely due to differences in firing rates between the conditions as similar to study 1a, participants who endorsed bombing rated their comfort as significantly lower ($M = 4.49$, $SD = 1.82$) than participants who did not endorse bombing ($M = 5.45$, $SD = 1.74$), $U(N_{Endorse} = 94, N_{Not\ endorse} = 177) = 5681$, $p < .001$, $r = 0.32$.

When asked to rate how much the ISIS fighter and the bystander each deserved to die, participants were again high in condemnation of the target ISIS soldier ($M = 5.43$, $SD = 1.48$), an effect which did not differ by condition, $\chi^2(6) = 5.19$, $p = .520$, $\varepsilon^2 = 0.02$. However, as in Study 1a, there was a strong impact of condition on ratings of whether the bystander deserved to die, $\chi^2(6) = 66.37$, $p < .001$, $\varepsilon^2 = 0.25$. Pairwise comparisons showed the unidentified bystander condition ($M = 2.95$, $SD = 1.67$) was significantly lower than the ISIS soldier condition ($M = 4.41$, $SD = 1.83$, $W = 5.11$, $p = .006$), and significantly higher than the conflicted Muslim ($M = 1.89$, $SD = 1.47$, $W = -4.26$, $p = .041$), the innocent bystander ($M = 1.88$, $SD = 1.62$, $W = -4.53$, $p = .023$), the anti-ISIS Muslim ($M = 1.76$, $SD = 1.45$, $W = -5.04$, $p = .007$). The unidentified bystander condition was marginally higher than the extremist Muslim ($M = 1.95$, $SD = 1.64$, $W = -4.17$, $p = .050$), and no different from the war profiteer bystander ($M = 1.94$, $SD = 1.47$, $W = -3.80$, $p = .102$). These findings largely mirror the findings of Study 1a.

**Qualitative justifications.**   We asked participants to justify their bombing decision with a free response of a few sentences. Participant answers were grouped by the bystander's identity and the participant's decision to fire. A coding scheme was developed post hoc that included 14 categories (S4 Table in S1 File). Across conditions, the most common type of justification for endorsing the bombing was *The greater good* ($N = 54$ out of 94, 57%) which included statements like "The death of the innocent goat farmer, however tragic, is one death compared to the possible hundreds of deaths that could result from the manufacture of chemical weapons from the ISIS operative also killed in the missile attack. One innocent's death compared to the death of hundreds of innocents," and "I would personally hate myself for pulling the trigger and killing both people, but in this scenario it's the one for the many. If the chemical weapons guy gets to make chemicals it could lead to greater loss of life. Not happy about my decision, but that's the one that likely has to be made as is war." For participants who did not endorse the bombing the most common kind of justification was *Appeal to innocence* ($N = 124$ out of 177, 70%) which included statements like "He should not—would never condone the killing of an innocent person," and "In America we do not believe in killing innocent people." Because in the majority of our conditions the bystander held some degree of innocence in the conflict, this kind of justification being so common is reasonable.

**Justifying bombing the unidentified bystander.**   Study 1a showed participants were surprisingly accepting of the bombing that would kill the unidentified bystander, so we looked specifically at the justifications of participants who opted to fire in the unidentified bystander condition. In the unidentified condition 18 out of 37 (49%) chose to fire. Of those, 8 (44%) cited *The greater good* as a justification with answers like "Can't afford to risk the people being able to make more weapons," and "If the pilot doesn't fire the missile and kill the operative then many more than just one other innocent person may die." This is reflective of the data overall across conditions. However, unique to this condition were 5 (28%) participants who justified the bombing on the assumption that the bystander was part of ISIS including statements like "I said yes because more than likely he ran into someone he knew at home (sic) so they probably are with him making them more than likely ISIS also," and "Chances are that the person the operative ran into the house with is also an operative of ISIS." This sample is quite small, but suggests that ambiguity about the identity of a bystander may lead around a quarter of participants to assume they are an enemy and endorse a bombing they otherwise may have not.

## Discussion

Overall, the main finding that endorsement of the bombing was much higher for the unidentified bystander compared to the innocent bystander was replicated from Study 1a. This suggests the finding was not dependent on public sentiment at the height of the armed conflict in 2016 and extended into 2020. Additionally, exploratory analysis into how participants justified their decision showed that around a quarter (28%) of participants in the unidentified bystander condition justified the bombing because of the possibility or the assumption that the unidentified bystander was an enemy combatant rather than an innocent civilian. It remains unclear to what extent this assumption impacts participant's reasoning about the dilemma, so in a follow-up study we asked directly.

## Study 1c

When bystanders to a military target are unidentified, there is a possibility they are a civilian, but also a possibility they are another enemy combatant. Studies 1a and 1b showed participants endorse bombing an unidentified bystander at rates much higher than an innocent. Many participants even reported they thought the unidentified bystander deserved to die. Furthermore, when

given a chance to explain their decision, around a quarter of participants who endorsed bombing the unidentified bystander believed the bystander was likely to be a combatant rather than a civilian. A possible explanation for the high rate of bombing the unidentified bystander is a tendency to assume they are an enemy. To investigate how participants are thinking about unidentified bystanders we asked directly. Given participants have no information about this person, if participants are unbiased, we would expect no consistent pattern of answers, i.e., the average response should be around the midpoint. If, however, participants tend to assume guilt, we would expect significantly more "yes" answers and a higher probability the bystander is an enemy, especially for those who endorse the bombing. Preregistered: https://aspredicted.org/TPK_HTT

## Method

To better understand the responses to the unidentified bystander condition from Studies 1a and 1b, we ran a small-scale replication of only the unidentified bystander condition ($N = 93$) as part of a larger study run in 2021 and not reported here. We asked questions identical to Study 1a and added two added additional questions: "The pilot followed the ISIS operative to a farmhouse where a second person already was. Do you think this second person is another member of ISIS?" (*Yes/No*) and "Please rate how likely it is the second person is another ISIS member" on a scale from 1 (*Extremely unlikely*) to 100 (*Extremely likely*). The former, binary measure was analyzed with a proportion test while the latter continuous measure was analyzed with a one sample $t$ test. For a proportion test, a sensitivity analysis showed 80% power to detect effect sizes of $h = 0.29$. For a one sample $t$ test, a sensitivity analysis showed 80% power to detect effect sizes of *Cohen's d* = 0.29.

## Results

In a replication of the unidentified bystander condition ($N = 93$), when asked, a majority assumed the unidentified bystander was part of ISIS (61%, significantly >50%, $p = .038$, 95%CI [60.62%, 71.22%]), with an average probability judgment above the midpoint, $M = 57.60$, $SD = 23.52$, student's $t(92) = 3.12$, $p = .002$, 95%CI [52.76, 62.45], $d = 0.32$. Participants who endorsed bombing gave a higher percent chance the bystander was a combatant ($M = 70.89$, $SD = 17.45$) than those who refused to bomb ($M = 45.67$, $SD = 21.92$), $t(91) = -6.09$, $p < .001$, $M_{Diff} = -25.21$, $SE_{Diff} = 4.14$, 95%CI [-33.44,-16.99], $d = -1.26$. These results demonstrate that many people believe an unidentified bystander to be an enemy combatant, despite no evidence this may be the case. In the absence of identity information, participants assuming the bystander was an enemy combatant felt free to endorse the bombing. This pattern suggests either motivated justification of bombing, or increased willingness to bomb given assumptions of enemyship.

## Discussion

Participates were significantly more likely to assume the unidentified bystander is an enemy rather than a civilian. Participants may endorse firing because of this assumption about a completely unknown person in the warzone. Perhaps the mere presence in a warzone is taken to suggest that unidentified individuals must be enemies—a dangerous assumption. To test this, Study 2 directly manipulated the probability the unidentified target was civilian and measured endorsement of bombing.

## Study 2

In Study 1, participants frequently endorsed sacrificing an unknown bystander to kill an enemy, many assuming the bystander was also an enemy. Different people may have differing

prior beliefs about the chances of a random person in a warzone being a combatant. Therefore, we directly manipulated the likelihood of the unknown bystander being a civilian in the vignette by giving a precise percentage that the bystander was an enemy or civilian. We again measured participant's endorsement of the bombing with the same measured used in Study 1a. Preregistered: asdpredicted.org/RDM_ABG.

## Method

Participants were 309 adult American MTurkers, with twelve (3.9%) excluded for failing a competence check, leaving 297 (Table 1). We aimed for >250 participants, a minimum for stable correlations [31]. A sensitivity analysis using G*Power 3 [32] showed 80% power to detect effect sizes of $OR = 0.66$ in a binary logistic regression. Participants read the unidentified bystander vignette from Study 1, except we manipulated the percentage likelihood the bystander was a member of ISIS from 0% to 100% in 10% increments (e.g., "there is a 70% chance they are ISIS, 30% chance they are a civilian"). Additionally, we changed the ISIS target from a chemical weapons maker to an ISIS leader (see S1 File for full text). We predicted that participants would rate sacrifice increasingly acceptable as the chance they are ISIS increased, but a substantial number (50%) would sacrifice even at maximal uncertainty. As in Study 1, we asked about firing endorsement as a binary yes/no question and as a 7-point Likert item for the acceptability of firing. We also asked the same manipulation check questions, with one change: instead of asking how deserving of death each person was, we asked how appropriate each target would be. Specifically, "If the [ISIS leader/unknown person] was alone in the farmhouse, would they be an appropriate target for the pilot to fire on?" with a Likert response between 1 (*Not at all appropriate*) and 7 (*Very appropriate*).

## Results

When asked whether or not they endorsed the bombing, people were more willing to bomb as the likelihood the bystander was ISIS increased in a binary logistic regression model, odds ratio $\chi^2(1, N = 297) = 29.5$, $p < .001$, $(OR) = 1.02$, 95% CI[1.01,1.03], McFadden's $R^2 = .07$ (Fig 4). When probability was 50%, close to half (40.0%) endorsed firing (95% CI[21.1, 61.3]); at 40% probability a minority endorsed firing (25.9%, 95% CI[11.1, 46.3]). This pattern shows

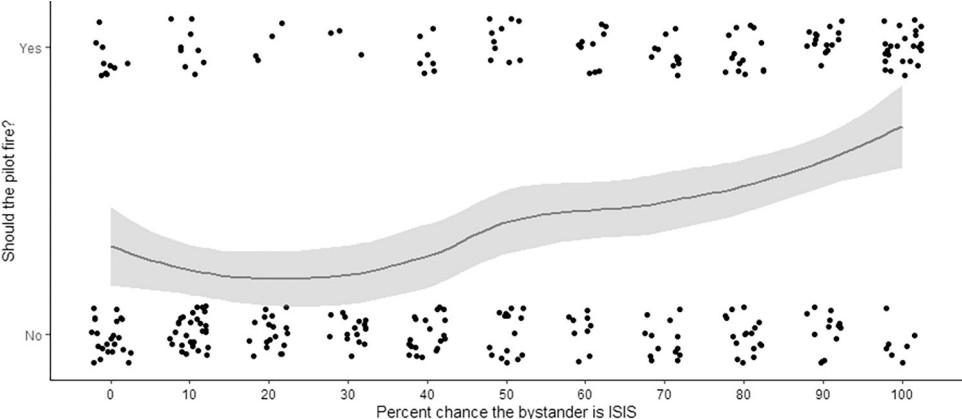

**Fig 4. Firing endorsement by percent chance the bystander is also in ISIS, scatterplot with applied jitter, Study 2.**
Participants were given a percent chance the bystander is an enemy combatant (ISIS member) from 0% to 100% and asked "should the pilot fire? [yes/no]. At 50% when there is an equal chance the bystander is an enemy versus a civilian, 44% of participants say the pilot should fire. This proportion of "yes" answers increases linearly as the chance the bystander is ISIS increases above 50%. Includes a loess line with grey area indicating confidence region.

even under uncertainty, many people still choose to bomb at rates similar to Study 1a. Additionally, at 100% probability the bystander was a civilian, nonetheless 30.3% agreed to bomb, comparable to the innocent civilian condition in Study 1a (23.1%). When rating the acceptability of firing on a 1–7 scale, bombing was similarly rated more acceptable as the likelihood the bystander was ISIS increased in a linear regression $F(1,295) = 25.4$, $p < .001$, $R^2 = 0.08$. Consistent with Study 1a, people who endorsed firing were less comfortable with their decision ($M = 4.47$, $SD = 1.90$) than people who did not ($M = 5.48$, $SD = 1.69$), $U(N_{Fire} = 117, N_{Don't\ Fire} = 180) = 7187.50$, $p < .001$, *biserial rank correlation* = 0.32.

Participants also rated how appropriate it would be to bomb each individual, the ISIS soldier and the bystander, if they were alone. We computed a linear regression depending on percent chance the bystander is in ISIS. As expected, for the ISIS leader there was no effect of condition, $F(1,295) = 0.88$, $p = 0.348$, $R^2 = .003$, but participants rated the bystander as a more appropriate target to fire upon as the percent chance they are ISIS increased, $F(1,195) = 22.97$, $p < .001$, $R^2 = .072$. Overall, participants rated the ISIS leader as a highly appropriate target ($M = 6.02$, $SD = 1.46$) and rated the bystander as a less appropriate target ($M = 1.87$, $SD = 1.47$).

## Discussion

These results supported both hypotheses. As the probability that an unidentified bystander was an enemy increased, so did willingness to kill them; yet, under the highest uncertainty (50% chance the bystander was ISIS), almost half of participants endorsed bombing. Hence, many participants appear quite willing to risk sacrificing an innocent civilian to kill an enemy. Such findings raise the question of mechanism. Perhaps this effect reflects motivated reasoning by American participants due to their position in the conflict and their reasonable bias against ISIS; if so, then a different pattern should emerge for a fictional conflict that avoids motivation to support one side over the other.

## Study 3

We examined whether participants would be less willing to sacrifice innocent bystanders in a fictional conflict. Studies 1 and 2 asked Americans about a real-world conflict between the US and ISIS, where motivated reasoning favoring the ingroup may drive assumptions, limiting generalizability. To test this possibility, we manipulated whether the conflict was between America and ISIS or two fictional countries, where motivation and ingroup affiliation should not matter. Specifically, in the real-world conflict condition we replicated the unidentified condition from Study 1 regarding an ISIS fighter and unknown bystander; in the fictional conflict condition participants read about an identical situation about a pilot from "Nibia" contemplating a strike against "Sorovia Federation fighters" (see S1 File for full text). Prior studies described the target of the attack as an ISIS soldier known for making chemical weapons; we worried this detail could make readers biased against the target in the fictional conflict condition since chemical weapon use is widely seen as taboo in modern warfare. For this reason, both conditions in Study 3 describe the target more neutrally as "a prominent leader and military strategist for [ISIS/the Federation]." We preregistered the study: aspredicted.org/QFW_ZFZ.

## Method

We recruited 223 adult American MTurkers, excluding 20 (9%) for failing a competence check, leaving 203. The pwr package for R revealed 80% power to detect Cohen's $w = 0.20$. The *real-world conflict* condition replicated the Study 1 unidentified bystander condition. The *fictional conflict* condition was identical except involving fictional countries *Nibia* and *Sorovian*

*Federation* (S1 Table in S1 File). Participants reported bombing endorsement as in Study 1a, and rated bombing acceptability 1 = *not at all*, 7 = *very much*. We described the main target as *leader* rather than *chemical weapons maker* so as to not vilify one side of the conflict. Participants also reported the same manipulation check questions as Study 2. As in Study 1c, participants also reported whether they thought the bystander was an enemy combatant or a civilian, both as a binary yes/no and as a continuous likelihood. For the real-world conflict condition these were the same as Study 1c and the wording was adapted for the fictional conflict condition, e.g. "The pilot followed the Sorovia Federation leader to a farmhouse where a second person already was. Do you think this second person is another member of the Sorovia Federation?".

## Results

The real-fictional manipulation revealed no significant effect for the binary choice of bombing or not, $\chi^2(1,203) = 2.39$, $p = 0.122$, $w = 0.11$. Next, we tested whether the two conditions were significantly *equivalent* via a two-sided equivalence test [33]. The TOST procedure for two proportions, with equivalence bounds of the raw score of $\Delta_L = $ -0.1 and $\Delta_U = 0.1$ (a 10% change in proportion who fired) revealed that the two conditions were not statistically equivalent, as the larger of the two *p* values is greater than 0.05, $z = .03$, $p = .511$. As both tests were nonconclusive, we ran similar analyses on the continuous measure of bombing acceptability. As with the binary measure of bombing endorsement, the Likert scale bombing acceptability measure also showed no difference between conditions, $t(201) = 0.91$, $p = 0.363$. We ran a TOST procedure for an independent samples *t*-test with unequal variances with equivalence bounds of $\Delta_L = $ -0.7 and $\Delta_U = 0.7$ (a 10% change in ratings of acceptability). This test suggested that the two groups were statistically equivalent, because the larger *p* value was still less than .05, $t(196.2) = $ -4.07, $p = < .001$.

As in prior studies, participants who read the real-world war vignette and those who read the fictional war vignette were no different in their comfort with their decision, $t(201) = $ -.35, $p = 0.727$. As in previous studies, across groups, participants who opted to fire showed less comfort with their decision (M = 4.27, SD = 1.78) than those who opted not to fire (M = 5.20, SD = 1.77), $t(201) = $ -3.48, $p < .001$. Participants did not differ between conditions in judgments of the appropriateness of bombing the enemy leader, *Welch's* $t(200.56) = 1.29$, $p = 0.200$, or the bystander, *Welch's* $t(194.09) = 0.04$, $p = 0.971$. However, as expected, participants judged the enemy leader as a significantly more appropriate target for firing a missile at (M = 5.78, SD = 1.57) compared to the bystander (M = 1.72, SD = 1.41), $t(202) = 27.78$, $p < .001$.

When asked whether they thought the bystander was a combatant, there was no difference between participants in the real world conflict condition (48.51%, not different from 50%, $p = .842$) and the fictional conflict condition (50.00%), $\chi^2(1,203) = 0.04$, $p = .832$. Similarly, when asked to give a likelihood that the bystander was a combatant, there was no difference between the real-world condition (M = 55.81, SD = 21.95) and the fictional condition (M = 56.69, SD = 20.60), $U(N_{real-world} = 101, N_{fictional} = 102) = 4973$, $p = .670$. The average likelihood the bystander was a combatant was significantly above 50% in both the real world condition, $W(101) = 2326$, $p = .008$, 95%CI[52.00,63.00], rank biserial correlation = -.08, and the fictional condition, $W(102) = 2276$, $p = .002$, 95%CI[53.00,65.00], rank biserial correlation = -.13.

## Discussion

Whether participants considered a conflict between the US and ISIS or two fictional countries, a similar proportion endorsed killing an unknown bystander to bomb a military target. This

pattern emerged on both the dichotomous measure and a more sensitive continuous measure with additional power to detect significant similarity. Hence, results do not seem to reflect motivated reasoning predicated on participant's position in the conflict; rather, they appear to reflect a general tendency to assume that unidentified bystanders are likely enemy combatants.

Participants rated the bystander as more likely to be a combatant than a civilian in both the real world and fictional conditions when asked to give a percentage. This is similar to the finding in Study 1c and shows that the tendency to assume guilt over innocence on the battlefield is not confined to the conflict between the US and ISIS. Though significant, this difference was small, and when reporting the binary yes/no measure of whether they thought the bystander was a combatant, participants in both conditions gave answers around 50%, contrary to Study 1c which showed a strong majority favoring combatant. However, both measures showed the pattern of judgment did not change whether participants judged a US ISIS conflict or a conflict between fictional countries.

The high rate of bombing unidentified bystanders may seem an error in judgment. Yet, from a "total war" perspective, treating civilians as part of a global struggle between military powers, harming civilians may seem like a rational trade-off. Next, we examined how participants' attitudes toward total war impacts their decisions.

## Study 4

In Studies 1–3, participants considered bombings that would kill an enemy and a bystander. Bystanders varied in relationship to enemies but were always described as locals to the combat region. Perhaps the 20–30% of participants endorsing bombing innocent bystanders viewed them as not 'wholly' innocent: rather people may infer enemyship from the target's mere presence in the region. Such beliefs may reflect "total war" beliefs that warfare involves a struggle between nations that extends beyond military combatants to civilian populations who contribute indirectly to conflict.

If so, then participants may be more willing to sacrifice innocent local bystanders (i.e., Iraqis) than members of neutral foreign nations (e.g., Sweden), as the latter cannot be construed as an enemy combatant even under total war beliefs. Crucially, this effect should pertain only for innocent targets, not those who join ISIS voluntarily (whether foreign or local). Study 4 therefore manipulated bystander nationality (Iraqi vs Swede) and innocence (documenting ISIS vs aiding ISIS). We also developed an exploratory measure of support for total war using a novel questionnaire. We used this to assess whether total war beliefs predict bombing endorsement; participants higher in such beliefs should be more willing to bomb, especially for local vs foreign targets, preregistered: https://aspredicted.org/M8G_MGC.

### Method

We recruited 602 American participants through Prolific, excluding 23 (3.8%) for failing a competence check, leaving 579. Participants ranged in age from 18 to 84 ($M = 38.46$, $SD = 14.43$). The pwr package for R showed 80% power to detect Cohen's $w = 0.14$. The study used four versions of the Study 1a vignette, manipulating bystander identity and guilt in a 2 (foreign vs local) by 2 (civilian vs ISIS affiliated) design. The bystander was described as either "a local Iraqi reporter" or "a foreign reporter from Sweden," who either writes articles "informing the world about ISIS activity," or "in support of ISIS and their ideology." Participants answered the same questions as in Study 1a, excluding the comfort with decision question.

As an exploratory measure, we developed ten total war belief questions (see S1 File) to assess how much participants consider all citizens and infrastructure in conflict areas legitimate targets for attack. The measure consisted of ten statements for which participants marked

their agreement. For example, we asked, "During war it is acceptable to bomb cities and other population centers in an enemy nation if it results in a crucial strategic advantage to help end the war" and "In war, everything and anything is fair game." (1 = *strongly disagree*—7 = *strongly agree*, α = 0.92, see S1 File). An average of these ten statements was taken as a score of total war attitudes. We predicted that people scoring higher on this measure would endorse bombing more often, especially for local versus foreign targets.

## Results

**Endorsing bombing.** A generalized linear model predicting bombing depending on bystander guilt and nationality showed that only guilt predicted firing rates, $\chi^2(1,579)$ = 133.30, $p < .001$, $w = 0.48$ (Fig 5). Neither nationality, $\chi^2(1,579) = 0.55$, $p = .46$, $w = 0.03$, nor the interaction were significant, $\chi^2(1,579) = 0.13$, $p = .72$, $w = 0.01$. Planned contrasts of Bonferroni corrected pairwise comparisons in a generalized linear model using a log link transform likewise showed sensitivity to bystander guilt, but not nationality. Specifically, we found significant differences between the following conditions: foreign-guilty vs. foreign-innocent

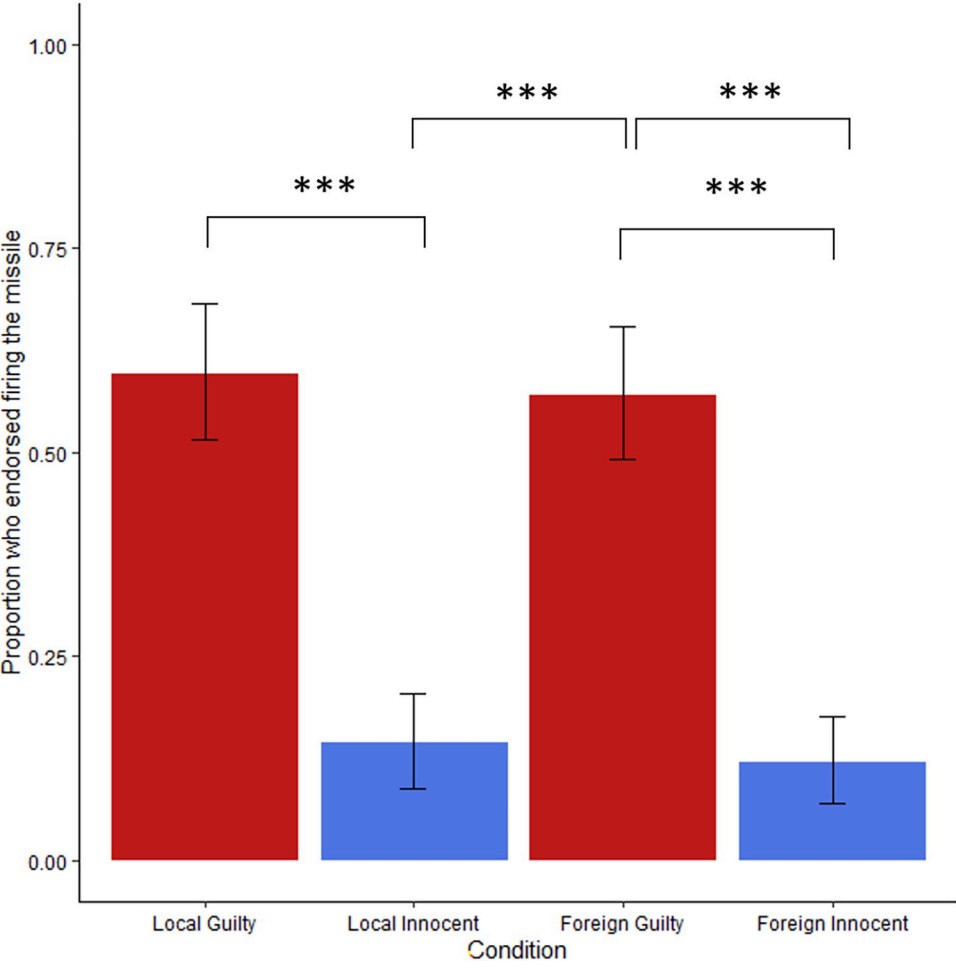

**Fig 5. The percentage of participants willing to fire depending on whether the bystander is innocent vs. Guilty and foreign (Swedish) vs. Local (Iraqi), Study 4.** Error bars reflect 95% CIs. Asterisks denote Bonferroni corrected p-values compared with the unidentified bystander condition (\*\*\*$p < .001$). There is a significant increase in firing when the bystander is guilty rather than innocent and no impact when they are foreign rather than local.

($Exp(B)$ = 0.11, $SE$ = .03, $p$ < .001), foreign-guilty vs. local innocent ($Exp(B)$ = 0.13, $SE$ = .04, $p$ < .001), foreign-innocent vs. local-guilty ($Exp(B)$ = 10.59, $SE$ = 3.19, $p$ < .001), and local-guilty vs. local-innocent ($Exp(B)$ = 0.12, $SE$ = .03, $p$ < .001). However, there were no differences between the foreign-guilty vs. local-guilty ($Exp(B)$ = 1.11, $SE$ = .27, $p$ = 1.00) or foreign-innocent vs. local-innocent conditions ($Exp(B)$ = 1.22, $SE$ = .42, $p$ = 1.00). Hence, bystander nationality did not influence bombing; all significant effects were driven only by bystander affiliation with ISIS. Interestingly, older participants were more likely to endorse bombing than younger participants in a logistic regression, though the effect size is very small, $\chi^2(1,579)$ = 4.60, $p$ = .032, $OR$ = 1.01, 95% CI[1.00,1.03]. This effect of age becomes nonsignificant when controlling for total war attitudes, see below.

**Acceptability of bombing.**   Participants reported "*How acceptable is it for the pilot to fire upon the building, killing both people inside*?" on a scale from 1 (*Not at all*) to 7 (*Very much*). A 2×2 between-subjects analysis of variance showed higher scores in the guilt vs innocence conditions, $F(1,567)$ = 140.03, $p$ < .001, $\eta^2$ = 0.20, but no significant difference between local vs foreign conditions, $F(1,567)$ = 0.36, $p$ = .551, $\eta^2$ = 0.00, and no significant interaction, $F(1,567)$ = 0.42, $p$ = 0.515, $\eta^2$ = 0.00. This pattern of continuous firing acceptability matched the pattern of dichotomous yes/no bombing endorsement.

**Deservingness to die.**   As a manipulation check of perceived bystander innocence vs guilt, participants rated how much both the bystander and the target (ISIS operative) deserved to die on scales from 1 (*Not at all*) to 7 (*Very much*). We computed a linear regression on bystander's deservingness to die depending on total war attitudes, guilt, and nationality, plus all interactions. The overall model was significant, $F(7,562)$ = 82.93, $p$ < .001, $R^2$ = .51. However, only the interaction between total war beliefs and bystander guilt emerged as significant, $t(579)$ = 5.25, $p$ < .001: when the bystander was guilty, people high in total war beliefs rated them more deserving of death than people low in total war beliefs. When the bystander was known to be innocent, total war beliefs had no impact on deservingness ratings (Fig 6). No other effects were significant (Table 2). As expected, participants consistently rated the ISIS operative high in deservingness to die with no difference between conditions, $F(3,318)$ = 0.77, $p$ = .51 ($M$ = 5.44, $SD$ = 1.60).

**Total war attitudes questionnaire.**   We conducted a principle components analysis using oblimin rotation with 500 iterations before convergence and 500 for rotation, retaining all

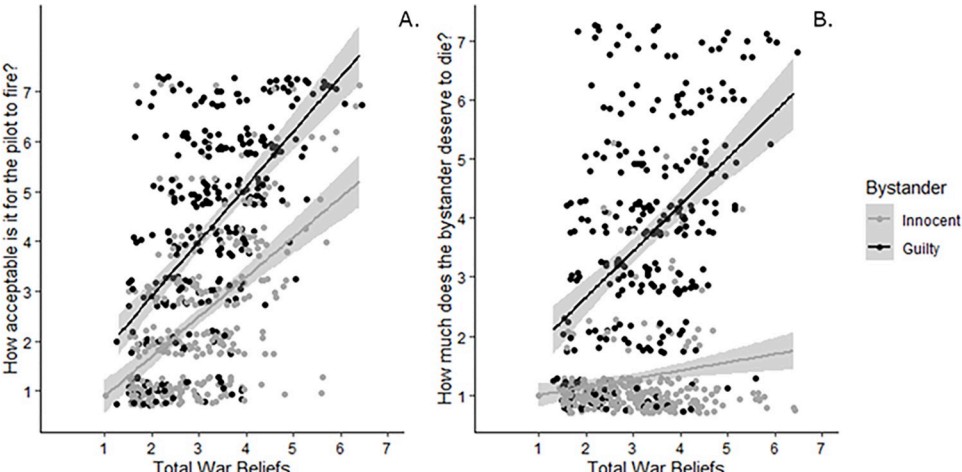

**Fig 6.** Firing acceptability ratings (Left) and ratings of how much the bystander deserves to die (Right) by total war beliefs and bystander's guilt, Study 4. Scatterplot with applied jitter includes linear regression line with grey area indicating confidence region.

**Table 2. Regressing ratings of how much the bystander deserves to die depending on their Guilt, nationality, and participant's support for total war, Study 4.**

| Predictor | Estimate | SE | t | p |
|---|---|---|---|---|
| Intercept [a] | 0.95790 | 0.31846 | 3.0080 | 0.0027 |
| Guilty (0 = Innocent, 1 = Guilty): | | | | |
| 1−0 | -0.27293 | 0.49038 | -0.5566 | 0.5780 |
| Nationality (0 = Iraqi, 1 = Swedish): | | | | |
| 1−0 | -0.22693 | 0.46627 | -0.4867 | 0.6267 |
| Total War Beliefs | 0.09340 | 0.09978 | 0.9360 | 0.3497 |
| Total War Beliefs ✳ Guilty: | | | | |
| Total War Beliefs ✳ (1−0) | 0.78179 | 0.14878 | 5.2548 | **< .0001** |
| Total War Beliefs ✳ Nationality: | | | | |
| Total War Beliefs ✳ (1−0) | 0.10334 | 0.14703 | 0.7028 | 0.4824 |
| Guilty ✳ Nationality: | | | | |
| (1−0) ✳ (1−0) | 1.00641 | 0.68739 | 1.4641 | 0.1437 |
| Total War Beliefs ✳ Guilty ✳ Nationality: | | | | |
| Total War Beliefs ✳ (1−0) ✳ (1−0) | -0.27870 | 0.20845 | -1.3370 | 0.1818 |

[a] Represents reference level; Bold indicates significance

factors with an eigenvalue greater than 1 [34]. Results showed a single factor with an eigenvalue of 5.96 accounting for 59.57% of the variance. Therefore, we treated these items as a single reliable measure ($\alpha$ = .92).

Next, we conducted a logistic regression on the decision to fire depending on total war attitudes, guilt, and nationality. The overall model was significant, $\chi^2(4,579) = 287.52$, $p < .001$, $w = 0.70$, with significant effects of total war attitudes, $\chi^2(1,579) = 152.91$, $p < .001$, $w = 0.51$, and guilt, $\chi^2(1,579) = 137.66$, $p < .001$, $w = 0.49$ (Fig 7), but not nationality, $\chi^2(1,579) = .50$, $p = .48$, $w = 0.03$, nor the interaction, $\chi^2(1,579) = .03$, $p = .87$, $w = 0.01$. By itself, age was a significant predictor of endorsing firing, however when it is included in this model alongside total war attitudes, it is not significant, $\chi^2(1,579) = 0.07$, $p = .798$, $OR = 1.00$, 95% CI[0.98,1.01]. It therefore makes sense that age is also correlated with total war attitudes, with older participants having higher belief in total war, *Pearson's r* = 0.20, p < .001, 95% CI[.12,0.27]. Any

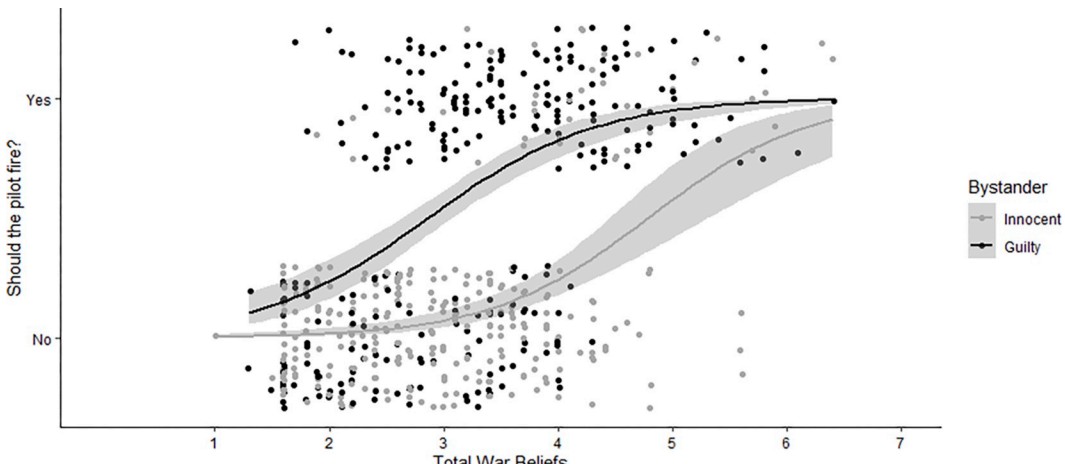

**Fig 7. Firing endorsement by bystander's guilt and total war beliefs, scatterplot with applied jitters Study 4.** Includes a loess line with a grey area indicating confidence region.

significant age effects on firing are therefore explained by older cohorts having higher support for total war.

The pattern was similar for scale ratings of bombing acceptability. We computed a linear regression on acceptability depending on total war attitudes, bystander guilt, and bystander nationality. The overall model was significant, $F(3,566) = 153.29$, p < .001, $R^2 = .45$, showing a small to moderate relationship (Table 2). Both total war attitudes $t(579) = 15.93$, β = .50, SE = .06, $p < .001$, and bystander guilt, $t(579) = 12.31$, β = .77, SE = .13, $p < .001$, but not bystander nationality, $t(579) = .62$, β = .04, SE = .13 $p = .54$, significantly predicted firing acceptability. We also ran a regression model including all 2- and 3-way interactions. This model was similar in predictive power to the above model, $F(3,562) = 67.32$, p < .001, $R^2 = 0.46$, and no interaction term was significant, so we report only the first model excluding interactions above [35].

## Discussion

This study clarified the role of bystander guilt, nationality, and total war beliefs on willingness to sacrifice bystanders to destroy enemies. We predicted support for total war would permit harming an innocent local Iraqi but not an innocent foreigner. Conversely, we expected that both local and foreign civilians who support the enemy would be perceived as equally available for harm.

However, results did not entirely support predictions. Consistent with prior studies, bombing rates were higher for guilty bystanders associated with ISIS than innocent bystanders reporting on ISIS. However, bombing rates were no different whether bystanders were local or foreign. Participants treated local and foreign enemies similarly—but also unexpectedly treated local and foreign innocent targets similarly, contrary to our predictions. Instead, total war attitudes predicted overall increased willingness to bomb, and especially high perceptions that guilty targets (foreign or local) deserved to die.

Hence, people who endorse total war appear to admit that some bystanders are innocent, yet nonetheless accept bombing them—they appear to view sacrificing innocents worthwhile to damage a known enemy. Intriguingly, it made no difference whether the innocent bystander to be sacrificed was local (thus presumably part total war conflict) or foreign (thus presumably not part). This pattern suggests that people who endorse total war beliefs may view any innocent targets as justified sacrifices in pursuit of damaging enemies, rather than only local civilians embroiled in total war.

It should be noted that participants were generally less willing to bomb both innocent targets in this study compared to Studies 1 and 2. It may be that this change reflects the edits to the scenario which clarified the conviction of the bystander's anti-ISIS stance (e.g., 'he does not support ISIS and never has'), or the description of his position as a reporter rather than a farmer, as this may have increased inferences that the innocent target is not merely neutral but possibly actively working against ISIS in the region. Alternatively, this pattern could reflect shifting public opinion on war in America, as this data was collected during the first days of the Russia-Ukraine conflict. Additionally, this study collected a sample from Prolific instead of mTurk, a platform with a different pool of users. Regardless, despite these changes to the paradigm, a substantial proportion of participants continued to endorse bombing regardless of the innocence of the bystander. It is an open question whether these findings are specific to US participants, so in a fifth study we replicated our method with a non-US sample.

## Study 5

Study 5 replicated Study 1 using United Kingdom participants. Pre-registered: https://aspredicted.org/W87_DB2.

## Method

We recruited 302 adult participants from the UK through Prolific. We excluded five for failing a competence check leaving a final sample of 297. The *pwr* package for R revealed 80% power to detect effect sizes of $w = 0.21$. In addition to items used in Study 1, we also included the total war attitudes questionnaire from Study 4, as well as the additional items for participants assigned to the unidentified bystander condition: "The pilot followed the ISIS operative to a farmhouse where a second person already was. Do you think this second person is another member of ISIS?" (*Yes/No*) and "Please rate how likely it is the second person is another ISIS member" on a scale from 1 (*Extremely unlikely*) to 100 (*Extremely likely*). If the effects we observed in previous studies are specific to American participants, we predict that using a sample from the UK will show differences.

## Results

As in Studies 1a and 1b, the bystander's identity significantly impacted bombing endorsement, using a logistic generalized linear model, $\chi^2(6,297) = 36.13$, $p < .001$, $w = 0.35$. However, bombing rates were generally lower than in the American sample. For example, only 32.5% of UK participants endorsed bombing the unidentified bystander, compared to 58.5% of Americans in Study 1 (Fig 8). Furthermore, UK participants did not bomb the unidentified bystander significantly more than other conditions. In addition to being asked whether the pilot should fire, our main dependent measure, participants were also asked "How acceptable is it for the pilot to fire upon the building, killing both people inside?" and answered on a Likert scale from 1 (*Not at all*) to 7 (*Very much*). A one-way between subjects analysis of variance showed significant differences between the vignettes across the report measures, $F(6,128.1) = 9.37$, $p < .001$. However, the unidentified bystander condition did not differ significantly from any condition except the ISIS soldier condition (Table 3). As in previous studies, the pattern of acceptability matched the pattern of endorsing firing.

However, UK participants in Study 5 replicated the pattern of American participants in Study 1 by rating the unidentified bystander as more deserving of death ($M = 2.19$, $SD = 1.34$, $N = 43$) than bystanders in all other conditions except a known enemy "ISIS soldier" condition, ($M = 1.34$, $SD = 0.89$, $N = 214$), which a Mann-Whitney U test showed was significant, $U$

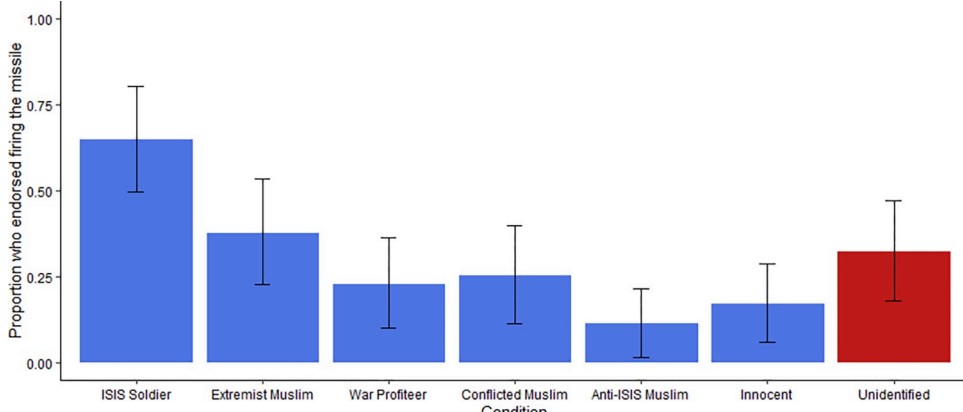

**Fig 8. Percent of "Yes" responses to "Should the pilot fire" depending on bystander identity, Study 5.** Error bars reflect 95% CIs. Asterisks denote Bonferroni corrected p-values compared with the unidentified bystander condition (*$p < .05$, **$p < .01$, ***$p < .001$). Unlike prior studies using American participants, bombing rates for the unidentified bystander condition did not differ significantly from any other condition.

**Table 3. Regressing ratings of how much the bystander deserves to die depending on bystander identity and total war beliefs, Study 5.**

| Predictor | Estimate | SE | t | p | Stand. Estimate |
|---|---|---|---|---|---|
| Intercept [a] | 1.25585 | 0.466 | 2.6968 | 0.007 | |
| Condition: | | | | | |
| Unidentified–Innocent | -0.77244 | 0.643 | -1.2016 | 0.231 | 0.60297 |
| War profiteer–Innocent | -0.69059 | 0.624 | -1.1064 | 0.270 | 0.12265 |
| Anti-ISIS Muslim–Innocent | -0.31175 | 0.640 | -0.4872 | 0.626 | -0.01591 |
| Extremist Muslim–Innocent | -1.05885 | 0.687 | -1.5416 | 0.124 | 0.17931 |
| ISIS soldier–Innocent | 1.66063 | 0.680 | 2.4429 | 0.015 | 1.93266 |
| Moderate Muslim–Innocent | -0.06990 | 0.624 | -0.1119 | 0.911 | 0.00976 |
| Total war attitudes | -0.00744 | 0.195 | -0.0382 | 0.970 | -0.00502 |
| Total war attitudes ✳ Condition: | | | | | |
| Total war attitudes ✳ (Unidentified–Innocent) | 0.66642 | 0.252 | 2.6476 | 0.009 | 0.44986 |
| Total war attitudes ✳ (War profiteer–Innocent) | 0.34641 | 0.250 | 1.3880 | 0.166 | 0.23384 |
| Total war attitudes ✳ (Anti-ISIS Muslim–Innocent) | 0.11369 | 0.267 | 0.4262 | 0.670 | 0.07675 |
| Total war attitudes ✳ (Extremist Muslim–Innocent) | 0.52589 | 0.261 | 2.0140 | 0.045 | 0.35500 |
| Total war attitudes ✳ (ISIS soldier–Innocent) | 0.50111 | 0.256 | 1.9579 | 0.051 | 0.33827 |
| Total war attitudes ✳ (Moderate Muslim–Innocent) | 0.03347 | 0.241 | 0.1387 | 0.890 | 0.02259 |
| **Model Fit Measures** | | | | | |
| | | | **Overall Model Test** | | |
| **R** | **R²** | **F** | **df1** | **df2** | **p** |
| 0.721 | 0.520 | 23.5 | 13 | 282 | < .001 |

[a] Represents reference level

($N_{unidentified}$ = 43, $N_{civilian}$ = 214) = 2720, $p$ < .0001, $_{MDiff}$ = -1.00, 95% CI[-1.00, -0.00], *rank biserial correlation* = 0.41 (Fig 9). Thus, although UK participants were more hesitant than Americans to endorse bombing targets in general, and unidentified targets in particular, they nonetheless demonstrated a similar pattern of suspicion toward an unknown bystander—even higher than towards a war profiteer (Table 3).

**Total war attitudes.** Participants reported total war attitudes as in Study 4. UK participants also showed high reliability with *Cronbach's* α = 0.88. A principle component analysis using oblim rotation based on eigenvalues greater than 1 showed 2 components with eigenvalues of 4.91 and 1.30 and accounting for 49% and 13% of the variance respectively. Thus we again treat the scale as a single factor to compare with findings from Study 4. We again conducted a logistic regression on bombing decisions depending on total war attitudes and bystander identity. Results showed significant main effects of total war attitudes, $\chi^2(1,297)$ = 88.09, $p$ < .001, $w$ = 0.54, and bystander identity, $\chi^2(1,297)$ = 24.66, $p$ < .001, $w$ = 0.29 but no interaction, $\chi^2(1,297)$ = 7.41, $p$ = .29, $w$ = 0.16. Consistent with Study 4, total war attitudes predicted overall bombing endorsement, regardless of bystander identity.

This pattern was not as strong for scale ratings of bombing acceptability where a linear regression which predicted bombing acceptability depending on total war attitudes and bystander identity. Results showed significant main effects of total war attitudes, $F(1,282)$ = 17.02, $p$ < .001, but no main effect of bystander identity, $F(6,282)$ = 1.70, $p$ = .122, nor the interaction, $F(6,282)$ = 1.41, $p$ = .209. It could be that total war attitudes drive decisions of whether bombing is acceptable, but are distinct from actually endorsing whether or not to bomb.

We also computed a linear regression on ratings of the bystander's deservingness to die depending on total war attitudes and bystander identity. The overall model was significant, *F*

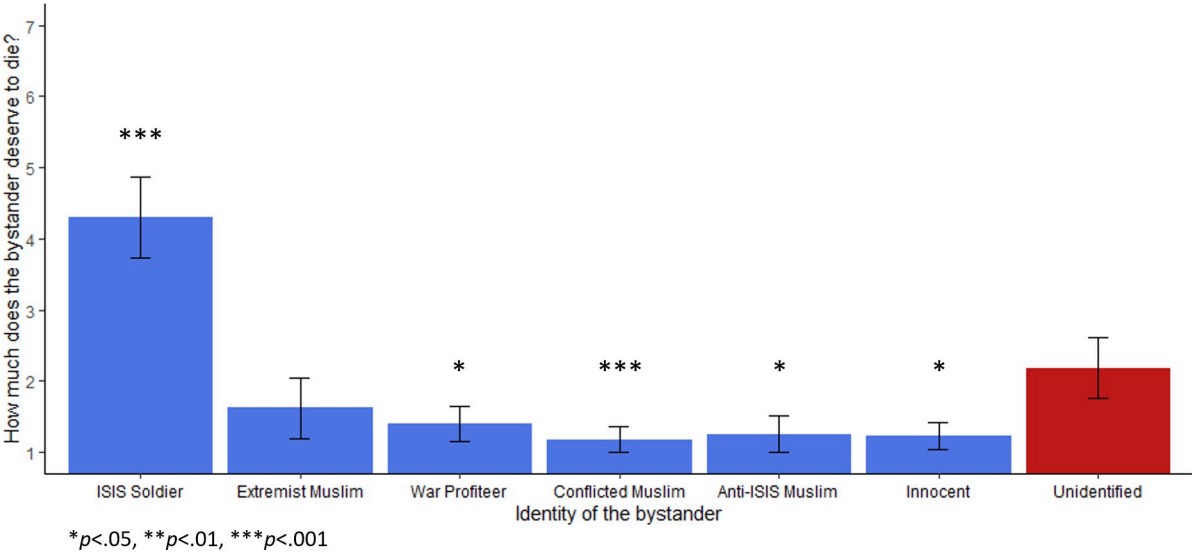

*p<.05, **p<.01, ***p<.001

**Fig 9. Ratings of bystander deservingness of death depending on their identity, Study 5.** Error bars represent a 95% confidence interval. Asterisks denote Games-Howell corrected p-values comparing with the unidentified bystander condition (*$p < .05$, **$p < .01$, ***$p < .001$). Participants answered the question "How much does the second person [the bystander] deserve to die?" with a Likert scale between 1 (*Not at all*) and 7 (*Very much*).

(13,282) = 23.47, p < .001, $R^2$ = .52 (Table 3). Similar to Study 4, there was a significant interaction between total war attitudes and bystander identity, $F(6,282) = 2.56$, $p = .020$. Total war beliefs were particularly related to judgments of how much some bystanders deserved to die but not others. Total war had little impact on judgments of the war profiteer, anti-ISIS Muslim, moderate Muslim, and ISIS soldier all of which were not significantly different from the innocent bystander; conversely total war beliefs had a greater impact on judgments of the unidentified bystander and the extremist Muslim (Fig 10).

Hence, similar to Study 4, total war beliefs seemed to especially increase perceptions that 'guilty' bystanders (closely associated to the enemy) deserved to die, rather than 'innocent' bystanders with less association. A possible exception was the ISIS soldier, which was only

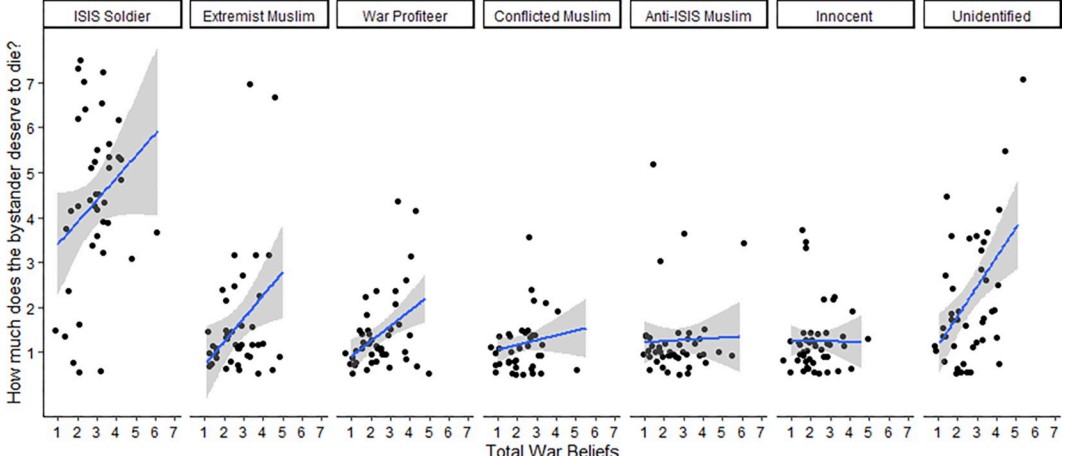

**Fig 10. Ratings of bystander deservingness of death depending on their identity, and total war beliefs, Study 5.** Includes linear regression line with grey area indicating 95% confidence region.

marginally ($p$ = .051) significantly different from the innocent bystander in how total war attitudes affected ratings of deservingness to die. Intriguingly, total war beliefs increased perceptions that the unidentified bystander deserved to die. This finding suggests that people higher in total war beliefs not only endorse the sacrifice of bystanders more frequently they may also tend to view unidentified bystanders as guiltier.

**Assumptions about the unidentified bystander.** We asked participants who read about the unidentified bystander ($N$ = 43) whether they believed he was part of ISIS, *yes* or *no*, and their probability estimate. In this UK sample, 37% said yes, not significantly different from 50% $p$ = .13, and substantially lower than the American replication of Study 1 where 67% (significantly greater than 50%) said yes. Likewise, the UK probability estimates were not different from 50% ($M$ = 52.4, $SD$ = 21.6), $t(42)$ = 0.73, $p$ = .471, 95% CI[45.76, 59.03], $d$ = 0.11, unlike American estimates which were significantly greater. Participants higher in support for total war were more likely to say "yes" when asked if the unidentified bystander was an ISIS member, $\chi^2(296)$ = 7.9, $p$ = .0049, $OR$ = 2.55, 95% CI[1.31, 5.63] and gave a higher probability the bystander was an ISIS member, $F(1,41)$ = 12.70, $p$ = .0009, $R^2$ = .24.

## Discussion

Replicating Study 1a with a UK sample produced different results from the US sample. Bombing endorsement was lower overall for UK participants, and the proportionally high rate of bombing of the unidentified bystander shown in Study 1 did not appear when using UK participants. Total war attitudes were lower in this UK sample than in the US sample which may explain the lower overall bombing rates. Additionally, UK participants did not display the same tendency to assume the unidentified bystander was more likely to be an enemy combatant than a civilian seen in Study 1. Assuming guilt when the bystander was unidentified was positively correlated to total war attitudes: lower support for total war among UK participants might be a factor underlying the disappearance of the high rate of bombing when the bystander is unidentified. This study suggests the findings from Studies 1–4 may be unique to US participants, possibly due to their relatively high support for total war compared to UK participants.

## General discussion

Five experiments examined how the identity of a wartime bystander influences willingness to sacrifice them to kill a dangerous enemy combatant. We discovered a potentially deadly tendency: when the bystander's identity was unknown, people tended to assume they were an enemy and therefore acceptable collateral damage. Crucially, ~50% of people across studies of American participants endorsed sacrificing unidentified bystanders despite no evidence they were enemies, a sacrificial rate higher than any identified target except a known enemy combatant (though not always significantly higher than some morally compromised targets). This effect emerged even when we explicitly provided probability estimates that the bystander was a civilian, emerged both during the height of the conflict (2016) and after (2020), and emerged for both real and fictional targets.

This bias toward sacrificing unknown bystanders appears to stem from assuming the unidentified person is an enemy. Our data suggest this finding is not merely due to ingroup bias—Americans supporting America's side of a war—as it emerged even when Americans judged a fictional war. Moreover, people endorsed sacrificing the bystander at rates of around 50% when the certainty they were an enemy was around 50%. Thus, consistent with other work (Watkins & Laham, 2019), wartime contexts may increase sacrificial acceptance, in part by allowing decision-makers to arrive at unflattering assumptions about unknown targets.

This pattern aligns broadly with research showing threatening contexts increase perceptions of harmful agents and outgroup categorization [18, 19].

Participants who endorse principles of total war such as "In war, everything and anything is fair game" were also more likely to endorse the bombing. We predicted individual differences in total war attitudes may lead to endorsing bombing when bystanders are civilians of an enemy nation rather than a friendly nation. However, this was not the case: higher support for total war principles predicted endorsing bombing generally and appeared insensitive to the bystander's nationality. When considering an innocent civilian bystander, participants who were high and those who were low on total war agreed that the bystander did not deserve to die; yet those high in total war attitudes were more likely to support a bombing that would kill that innocent civilian to also kill a dangerous ISIS member. This suggests supporters of total war are more likely to support sacrificing civilians as collateral damage, while still admitting that it is a sacrifice.

## Limitations

A replication using participants from the UK showed differences from US participants. Those from the UK were less likely to endorse bombings overall, were lower on support for total war, and were less likely to assume the unidentified bystander was a combatant compared to participants from the US. Although this may put limits on the generalizability of these findings, it is possible that low support for total war in the UK compared to the US is the driving factor for these differences as total war beliefs was positively related to both endorsing bombing and believing the bystander is likely to be a combatant. However, these relationships are correlational and future studies could benefit from more direct comparisons between countries across these measures and future work could generalize this paradigm to other nations and cultures [36].

Moreover, all studies recruited from the general population: trained military decision-makers could be either more or less hawkish in these decisions—evidence suggests both directions are possible. Research examining race-bias in police shootings finds trained police are less biased than civilians [37]. On the other hand, we find some evidence to suggest that those with ties to the military are *more* likely to endorse firing on the unidentified bystander (see S1 File: Study 1a Military Experience). Additionally, all studies wholly or in part referenced an ongoing real-world conflict, so it is possible shifting sentiment about this conflict may change the results of future studies. However, data from Study 1a was collected in 2016 during the conflict, and a 2020 replication (Study 1b) after the conflict reported similar levels of firing on unidentified bystanders.

All the studies reported here measure participants' willingness to endorse bombing a bystander they have no information about, and Study 1c shows the high rates in which participants infer the unidentified bystander is an enemy. However, one could argue that our unidentified bystander vignette *does* have circumstantial evidence of the bystander's affiliation with the enemy. Participants may reasonably assume the bystander is likely to be an enemy because of proximity to the enemy. Gestalt psychology [38] demonstrates that in general people are likely to view individuals who are close in proximity to be socially close as well. However, a study which replicated Study 1a while manipulating the bystander's proximity to the enemy showed no impact on endorsing the bombing (S1 Study in S1 File). Proximity to the enemy is unlikely to fully explain our participants' bombing endorsement since manipulating it directly had no impact on any dependent measures.

Choosing when to bomb bystanders in war is a choice informed by ideology and can be influenced by an individual's political beliefs. Aside from our measure of support for total war,

we did not collect data on political ideology and how this may affect bombing decisions. Future work could compare politics as a possible moderator to these effects.

## Conclusion

People often assume unidentified bystanders in a warzone are combatants and acceptable collateral damage. Rather than give bystanders the benefit of the doubt, people tend to treat them as "guilty until proven innocent." This could explain some part of the staggering cost of civilian life in modern armed conflicts around the world today. These findings have implications for military strategists who must decide whether to attack areas with enemy militants and unidentified bystanders. Our results support a common tendency in people to assume the bystanders are enemies, which can have deadly consequences if they turn out to be innocent civilians. The real-world cases of civilians struck by bombs could be the direct result of the very same error in judgment we report. To minimize civilian deaths, future research should investigate how to reduce this bias and get decision-makers to evaluate more carefully who their weapons are targeting.

## Supporting information

**S1 File. Supplemental materials.** Supplemental materials including S1-S4 Tables full text from all conditions in all studies, additional measures: Studies 1a, 1b, 1c, 2, qualitative coding categories for bombing justifications Study 1b, replications and pilot studies, total war questionnaire items, additional study, Also, S1 Fig. Rating of firing acceptability in Study 1a. Ratings are split by condition and range from 1 (Not at all) to 7 (Very Much). Error bars represent a 95% confidence interval. Asterisks denote Dwass-Steel-Critchlow-Fligner pairwise comparison p-values compared with the unidentified bystander condition (*p < .05, **p < .01, ***p < .001).
(PDF)

## Acknowledgments

We would like to thank Eli Zachowski, an undergraduate collaborator who contributed to the initial study design and implementation but was unable to be reached for authorship.

## Author Contributions

**Conceptualization:** Paul Conway, Andrew Vonasch.

**Data curation:** Scott Danielson.

**Formal analysis:** Scott Danielson, Paul Conway, Andrew Vonasch.

**Investigation:** Paul Conway, Andrew Vonasch.

**Methodology:** Scott Danielson, Paul Conway, Andrew Vonasch.

**Project administration:** Scott Danielson.

**Writing – original draft:** Scott Danielson, Paul Conway, Andrew Vonasch.

**Writing – review & editing:** Scott Danielson, Paul Conway, Andrew Vonasch.

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
