## [Decision Letter · Decision Letter 0]

16 May 2024

PONE-D-24-04279What I don’t know can hurt you: Collateral combat damage seems more acceptable when bystander victims are unidentifiedPLOS ONE

Dear Dr. Conway,

Thank you for submitting your manuscript to PLOS ONE. After careful consideration, we feel that it has merit but does not fully meet PLOS ONE’s publication criteria as it currently stands. Therefore, we invite you to submit a revised version of the manuscript that addresses the points raised during the review process.

**ACADEMIC EDITOR: **Please carefully review the comments and recommendations made by the two reviewers, and make revisions accordingly.==============================

We look forward to receiving your revised manuscript.

Kind regards,

Poowin Bunyavejchewin

Academic Editor

PLOS ONE

Journal Requirements:

Reviewers' comments:

Reviewer's Responses to Questions

**Comments to the Author**

1. Is the manuscript technically sound, and do the data support the conclusions?

Reviewer #1: Yes

Reviewer #2: Yes

2. Has the statistical analysis been performed appropriately and rigorously? 

Reviewer #1: Yes

Reviewer #2: Yes

3. Have the authors made all data underlying the findings in their manuscript fully available?

Reviewer #1: Yes

Reviewer #2: Yes

4. Is the manuscript presented in an intelligible fashion and written in standard English?

Reviewer #1: Yes

Reviewer #2: Yes

5. Review Comments to the Author

Reviewer #1: I am grateful for the for the opportunity to review the article "What I don't know can hurt you: Collateral combat damage seems more acceptable when bystander victims are unidentified". The research aim appears to address an underexplored area within its field, contributing potentially new insights.

The overall structure of the article is logical, with a clear progression from introduction to conclusion. The methodology section is detailed, providing a good understanding of the research design and analysis. The statistical analysis has been performed appropriately and rigorously. The results are presented clearly, with appropriate use of tables and figures. The researchers are aware of the limitations of their research.

The manuscript is well-structured as well as written and I only have minor suggestions. Perhaps it is worth outlining more clearly the gap that this research fills in the introduction, it may strengthen the fundamental argument of the article. The practical implications of the findings in the conclusion could further elevate the article's impact.

Reviewer #2: Lines 60-65 & Lines 70-73: What is the source of this information?

inclusion/exclusion: Were participants screened for their own ideas/prejudices regarding Islam and/or terrorism associated with Islam? Kindly clarify what kinds of efforts were made while including participants for the different experiments.

6. PLOS authors have the option to publish the peer review history of their article (what does this mean?). If published, this will include your full peer review and any attached files.

Reviewer #1: No

Reviewer #2: No

---

## [Author Response · Author response to Decision Letter 0]

1 Jul 2024

To the editors at PLOS ONE,

We thank you and the reviewers for their time and attention in reviewing this manuscript. Below addresses the changes made to the attached manuscript.

Response to Reviewer #1

 To address the suggestion outlining the gap the research fills we added 5 lines to the second to last paragraph of the introduction explicitly addressing the research gap (lines 82-87). We hope this clarifies the contribution of this paper.

 Along these same lines we also added a few points to the conclusion section of the general discussion to elaborate more on the practical implications of the research (lines 857-858, 862-863). This should make more obvious the meaning impact this research could have. We appreciate these suggestions and hope it will make the significance of this paper clear to the reader.

Response to Reviewer #2

As for the sources for lines 60-65 in the section about what Prospect Theory might predict about wartime judgments, we added two new citations (26 & 27) and two sentences of elaboration to further clarify (Lines 59-66).

For the source of lines 70-73, we have added a few lines of clarification that we are speculating about how belief in total war might predict wartime decision making. A thorough search of the empirical literature using multiple digital platforms found no empirical papers that assessed belief in total war as we describe it here. Thus, as to our knowledge, this is the first empirical paper to define and measure total war beliefs and how they apply to wartime decision-making. We make it clear that this is something that is unknown to set up how the findings of this paper fill this research gap (lines 74-77).

The reviewer also asks about inclusion or exclusion criteria about the participants’ ideas/prejudices regarding Islam and/or terrorism. We included everyone who took the survey so long as they gave complete data and passed competence check items. We did not measure or exclude participants based on prejudices of Islam or any other ideological or political beliefs. Though this is an interesting idea, and something we are considering in future studies, for an initial paper demonstrating our observed effects we wanted to cast as wide a net as possible. This was the case for all studies in the paper and we have added a note (lines 118-119) making that clear. This was also previously noted at the end of the discussion (lines 853-855).

 We also proofread for copy editing and made a few minor changes to grammar. These can also be found in tracked changes in the attached document. Thank you all for the feedback and we appreciate the suggestions at improving this manuscript. We hope these edits are to your satisfaction and we welcome any further feedback.

---

## [Decision Letter · Decision Letter 1]

4 Sep 2024

PONE-D-24-04279R1What I don’t know can hurt you: Collateral combat damage seems more acceptable when bystander victims are unidentifiedPLOS ONE

Dear Dr. Conway,

Thank you for submitting your manuscript to PLOS ONE. After careful consideration, we feel that it has merit but does not fully meet PLOS ONE’s publication criteria as it currently stands. Therefore, we invite you to submit a revised version of the manuscript that addresses the points raised during the review process.

**ACADEMIC EDITOR:  ** The Senior Editors have requested another round of review. Therefore, please review the comments made by the new reviewers and consider making the necessary revisions accordingly. I apologize for the delay in reaching a final decision. ==============================

We look forward to receiving your revised manuscript.

Kind regards,

Poowin Bunyavejchewin

Academic Editor

PLOS ONE

Journal Requirements:

Reviewers' comments:

Reviewer's Responses to Questions

**Comments to the Author**

1. If the authors have adequately addressed your comments raised in a previous round of review and you feel that this manuscript is now acceptable for publication, you may indicate that here to bypass the “Comments to the Author” section, enter your conflict of interest statement in the “Confidential to Editor” section, and submit your "Accept" recommendation.

Reviewer #3: All comments have been addressed

Reviewer #4: All comments have been addressed

2. Is the manuscript technically sound, and do the data support the conclusions?

Reviewer #3: Yes

Reviewer #4: Partly

3. Has the statistical analysis been performed appropriately and rigorously? 

Reviewer #3: Yes

Reviewer #4: Yes

4. Have the authors made all data underlying the findings in their manuscript fully available?

Reviewer #3: Yes

Reviewer #4: Yes

5. Is the manuscript presented in an intelligible fashion and written in standard English?

Reviewer #3: Yes

Reviewer #4: Yes

6. Review Comments to the Author

Reviewer #3: How did gender and cultural differences among participants affect their likelihood to endorse bombing in scenarios involving different types of bystanders, and what underlying factors might explain any observed disparities?

Reviewer #4: The article titled ‘What I don’t know can hurt you: Collateral combat damage seems more acceptable when bystander victims are unidentified” effectively explored the impact of war attitude on bystander victims. However, several points need to be addressed.

1. Approval for all studies should not come under 1st study.

2. As per the manuscript, “total war” belief in experiment is inferred if bystanders were viewed as not ‘wholly’ innocent. Is measurement of total war attitude objective and replicable?

3. No checks were performed concerning reliability and validity of the developed “total war belief” measure/questionnaire, and no standard method is reported for developing these questions. More details need to be provided, and it should be standardized before use.

7. PLOS authors have the option to publish the peer review history of their article (what does this mean?). If published, this will include your full peer review and any attached files.

Reviewer #3: **Yes: **Ilakkiya.L

Reviewer #4: No

---

## [Author Response · Author response to Decision Letter 1]

11 Sep 2024

To the editors at PLOS ONE,

We thank you and the reviewers for their time and attention in reviewing this manuscript. Below addresses the changes made to the attached manuscript. Any changes to the manuscript are highlighted using Track Changes.

Response to Reviewer #3

 We would like to thank you for your time looking over our manuscript. We have discussed your points and address them below. 

Now included in the first paragraph method are gender and age demographics for Study 1a (Lines 121-122) and Study 1b (Lines 235-236). Gender made no significant difference in predicting whether people endorsed the various bombings across conditions in Study 1a or 1b. We now make note of these null findings and the accompanying statistical test results for Study 1a (Line 183-187) and Study 1b (Line 242-243). Since these studies together total a sample of 735 participants and represent the largest breadth of situations that participants are shown across 7 conditions, we believe these findings sufficiently address the question of the impact of gender. It seems that either there is truly no effect of gender or effects are so small as to be negligible. As gender is not the focus of this paper we do not report further gender analyses in the remaining studies.

 Age similarly had no impact on firing rates in Studies 1a and 1b; we now note these findings and their accompanying statistics in Studies 1a (Line 186-187) and 1b (Line 243-244). Interestingly, Study 4 showed a small effect of age (Line 579-581), where older adults were more likely to endorse firing overall. This finding may reflect the parallel finding that older adults were more likely to score high on total war beliefs, since including both total war beliefs and age in a model as predictors of firing rendered age no longer significant (Line 622-627). In other words, Study 4 found that older participants were more likely to endorse bombing only because they are more likely to believe in the concept of total war than younger participants. We believe that incorporating these clarifications strengthens the paper. 

 Reviewer 3’s question on how culture impacts firing endorsement is well placed, as this is something we are also interested in pursuing in future research. We began to address it in Study 5 in this paper. Study 5 tested how American and British participants responded to the same vignettes. We found that Americans were more likely to support bombing than British participants. We suggest one cultural difference potentially accounting for this difference is cultural differences in total war beliefs (see Study 5). Study 5 in the paper addresses this question directly by testing how well findings generalize beyond an American sample. Although the predictive pattern was similar, suggesting that the relationships between the psychological mechanisms involved may be robust across culture, we nonetheless also report mean differences between the two populations—specifically, American participants were higher in total war beliefs than participants in the UK, and showed a corresponding increased willingness to bomb in line with theory. Hence, we not only report at least one important cultural difference in bombing willingness, but also suggest mechanism: that US versus UK differences in bombing endorsement can be explained by differential endorsement of total war. 

 That said, we acknowledge that thus far we have examined only two (fairly similar) cultures, and it would naturally be interesting to examine how well these patterns replicate across a variety of more dissimilar cultures, such as samples from China or the Middle East or the Global South. Alas, such investigations are beyond the scope of the current manuscript. In future studies we hope to investigate how factors like political ideology or social dominance orientation can predict how people judge situations like the ones we present in the current paper. However with 7 studies in the main manuscript and three supplemental studies (2 replications and one novel) we feel the paper is quite broad already and it would be better suited to explore additional questions in follow up papers. We write in the paper, “future studies could benefit from more direct comparisons between countries across these measures and future work could generalize this paradigm to other nations and cultures” (Lines 847-849). My co-authors and I would like to thank reviewer 3 for their suggestions and we feel it has improved the paper.

Response to Reviewer #4

We would like to thank you for your time looking over our manuscript. We have discussed your points and address each below. 

1. Approval for all studies should not come under 1st study.

The section about ethics approval of all studies is currently the last paragraph of the introduction (Lines 102-110). In case perhaps you meant the sample size and demographics Table 1, we have moved that to immediately follow this section at the end of the general introduction.

2. As per the manuscript, “total war” belief in experiment is inferred if bystanders were viewed as not ‘wholly’ innocent. Is measurement of total war attitude objective and replicable?

We appreciate the opportunity to clarify. The total war questionnaire is an objective measure, see supplemental materials Line 239 for full text of all items in the measure. Participants answered 10 Likert scale questions on a 1 to 7 scale ‘strongly disagree – strongly agree’. The average of these values was taken as a measure of total war beliefs (Line 562-563). 

The total war questionnaire is also reliable in its ability to consistently predict outcomes relevant to the construct in both studies where we used it (Study 4 Lines 746-778, Study 5 Lines 614-639). Thus, we conclude that the measurement is objective and replicable. To be sure, given the replication crisis any measure might be questionably replicable, but we used this measure in two high-powered studies and both showed strong internal consistency and similar effects so we have no more reason to doubt its replicability any more than any other scale used in psychological research. 

3. No checks were performed concerning reliability and validity of the developed “total war belief” measure/questionnaire, and no standard method is reported for developing these questions. More details need to be provided, and it should be standardized before use.

We are dedicated to ensuring our new questionnaire measures what it intends to. For this measure we wanted to know how much participants agreed with a certain mindset towards warfare. The concept of total war is one that has been used by historians for decades (Chickering, 1999, full reference in manuscript), but to our knowledge it has not been studied by empirical psychology. Rather than asking participants “do you believe total war is acceptable?” we used 10 questions asking about the core elements of this concept. Do each of these questions measure the same concept? We used two forms of statistical tests to demonstrate that they do.

The total war questionnaire passed two statistical measures of reliability. Cronbach’s alpha is a statistical measure of reliability with values between 0 and 1 with higher values being more reliable. For early-stage research it is recommended that measures have a Cronbach’s alpha of at least 0.7 to be considered reliable, and 0.8 for applied research (Nunnally & Bernstein, 1994). This measure replicated across Study 4 and Study 5 in its internal consistency with Cronbach’s alpha values well above the 0.7 and 0.8 cutoff. (Study 4 α=0.92, Lines 614-618, Study 5 α= 0.88, Lines 746-755). Thus, the total war questionnaire has statistically reliable internal consistency.

 Secondly, a principle component analysis was also reported for Study 4 (Lines 614-617) Study 5 (Lines 746-751). The results of these two tests indicate the internal consistency is good. Other components of reliability are test-retest reliability, which isn't relevant because we don't apply the measure in a repeated measures context, and inter-rater reliability, which isn't relevant because the measure doesn't involve subjective ratings. Questions were asked of all participants in an identical way in both studies 4 and studies 5. It is an objective scale. 

 In terms of validity, the items have strong face validity (see supplemental materials for full text of the items). We ask participants their agreement on ten statements relevant to the concept we intend to measure. Moreover, we believe content validity is high because the items of the scale measure a well-conceived concept. Study 4 and 5 both found the total war belief measure was a valid predictor of outcomes in ways suggesting it has good predictive validity. In both studies 4 and 5 high score on the total war questionnaire predicted endorsing the bombing in our specific vignettes. 

The total war questionnaire is also predicted by age, with older participants being more likely to endorse the idea of total war (Lines 623-629). This is particularly true for the age 65+ cohort, who would have grown up closer to a period in history where total war was widely supported in the US and UK. 

Finally, as noted in the manuscript (Line 556), the total war belief measure was used for exploratory purposes, so it was not necessary to conduct additional scale validation studies. The purpose of this article is not to establish the validity of this new measure, so doing so would go beyond the scope of this article. Nonetheless, there is a fair amount of evidence of the validity and reliability of the scale in the final two studies we present in this paper. 

We hope this discussion clarifies the validity and reliability of the total war beliefs measure. We note that in social psychology papers it is very common for researchers to develop new measures to capture constructs in an effort to clarify mechanism, and rarely do such measures come with detailed validation studies, which are more appropriate for papers focused on scale validation per se. We agree that it would be ideal to further develop and validate this measure, but doing so goes beyond the scope of the current paper. We hope this makes sense and thank the reviewer for the chance to clarify this measure. 

References

Nunnally, J. C.; Bernstein, I. H. (1994). Psychometric theory (3rd ed.). McGraw-Hill. ISBN 0-07-047849-X. OCLC 28221417

---

## [Editor Report · Decision Letter 2]

18 Sep 2024

What I don’t know can hurt you: Collateral combat damage seems more acceptable when bystander victims are unidentified

PONE-D-24-04279R2

Dear Dr. Conway,

We’re pleased to inform you that your manuscript has been judged scientifically suitable for publication and will be formally accepted for publication once it meets all outstanding technical requirements.

Kind regards,

Poowin Bunyavejchewin

Academic Editor

PLOS ONE
---

## [Editor Report · Acceptance letter]

26 Sep 2024

PONE-D-24-04279R2 

PLOS ONE

Dear Dr. Conway, 

I'm pleased to inform you that your manuscript has been deemed suitable for publication in PLOS ONE. Congratulations! Your manuscript is now being handed over to our production team.

Kind regards, 

on behalf of

Mr. Poowin Bunyavejchewin 

Academic Editor

PLOS ONE